# Occupational Health and Safety Measures in German Outpatient Care Services during the COVID-19 Pandemic: A Qualitative Study

**DOI:** 10.3390/ijerph18062987

**Published:** 2021-03-14

**Authors:** Mara Shirin Hetzmann, Natascha Mojtahedzadeh, Albert Nienhaus, Volker Harth, Stefanie Mache

**Affiliations:** 1Institute for Occupational and Maritime Medicine (ZfAM), University Medical Centre Hamburg-Eppendorf (UKE), Seewartenstr. 10, 20459 Hamburg, Germany; marashirin.hetzmann@justiz.hamburg.de (M.S.H.); n.mojtahedzadeh@uke.de (N.M.); harth@uke.de (V.H.); 2Department of Occupational Medicine, Hazardous Substances and Public Health, Institution for Statutory Accident Insurance and Prevention in the Health and Welfare Services (BGW), Pappelallee 33/35/37, 22089 Hamburg, Germany; a.nienhaus@uke.de; 3Institute for Health Service Research in Dermatology and Nursing (IVDP), Competence Centre for Epidemiology and Health Services Research for Healthcare Professionals (CVcare), University Medical Centre Hamburg-Eppendorf, Martinistr. 52, 20246 Hamburg, Germany

**Keywords:** outpatient care, COVID-19, occupational health and safety, health threats, needs

## Abstract

Due to the coronavirus disease 2019 (COVID-19) pandemic, outpatient caregivers are exposed to new serious health threats at work. To protect their health, effective occupational health and safety measures (OHSM) are necessary. Outpatient caregivers (*n* = 15) participated in semi-structured telephone interviews in May/June 2020 (1) to examine the pandemic-related OHSM that have been implemented in their outpatient care services, as well as (2) to identify their corresponding unmet needs. Interviews were recorded, transcribed and analysed by using qualitative content analysis in accordance with Mayring. Available OHSM in outpatient care services described by outpatient caregivers mainly included personal protective equipment (PPE) and surface disinfection means after an initial shortage in the first peak of the pandemic. Further OHSM implied social distancing, increased hygiene regulations and the provision of pandemic-related information by the employer, as well as the possibility to consult a company doctor. Our study revealed that OHSM were largely adapted to the health threats posed by COVID-19, however an optimum has not yet been achieved. There is still a need for improvement in the qualitative and quantitative supply of PPE, as well as on the organisational level, e.g., with regard to the development of pandemic plans or in work organisation.

## 1. Introduction

On 11 March 2020, the outbreak of the novel type of coronavirus was officially declared as a pandemic by the World Health Organization (WHO) [1,2]. By 3 January 2021, more than 1.7 million coronavirus positive cases have been confirmed in Germany so far and still the tendency is rising [3]. In times of a pandemic, the protection of healthcare professionals is vital for the maintenance of essential health services [4,5]. However, due to the rapid and sudden outbreak of the coronavirus disease 2019 (COVID-19) pandemic, new occupational risks, e.g., a high risk of infection, increased workloads and corresponding stress levels, [6,7], arose especially in healthcare professions that now have to be addressed. Healthcare professionals are particularly vulnerable to the health risks posed by the COVID-19 pandemic because they are in the front line of efforts to combat the outbreak [8,9,10]. Consequently, a considerable number of healthcare professionals are among those infected, similar to the severe acute respiratory syndrome (SARS) pandemic in 2003 [11]. At the beginning of January 2021, 45,992 healthcare professionals tested positive for COVID-19 according to § 23 “Infektionsschutzgesetz IfSG” (German Infection Protection Act) [3]. Besides hospitals, outpatient surgery services, preventive or rehabilitation services, § 23 also includes outpatient care services. Despite the serious health risks of COVID-19, among them, the development of pneumonia or in very severe cases lung failure and even death, [12] outpatient caregivers need to continue to care for the significant number of outpatient caregivers (980,000 [13]), even if precautions such as contact bans apply [14]. Aggravating the situation is the fact that certain groups in society are at higher risk for a severe course of a coronavirus infection [15], these at-risk groups are also present among healthcare professionals [6,11]. Therefore, the high figures of infection suggest that outpatient caregivers are working in a high-risk area for health during this times. Hence, any potential health hazards in their workplace caused by the pandemic are important and should be prevented [15]. To protect the health of outpatient caregivers during the COVID-19 pandemic, appropriate and targeted occupational health and safety measures (OHSM) are required, taking into account the current circumstances affecting the safety and health of employees at work (ArbSchG §§ 3,4). In doing so, necessary OHSM must be implemented, taking into account the circumstances affecting the safety and health of employees at work. When implementing OHSM in a setting, the hierarchy of hazard control from technical to organisational to personal protective measures must be considered [16]. In addition, the effectiveness of the measures has to be reviewed and, if necessary, adapted to changing circumstances. The great relevance of OHSM to prepare workplaces in the healthcare sector to deal with COVID-19 is also reflected in the fact that COVID-19 infections in health care professions are recognised as an occupational disease [17]. The experiences gained on OHSM in paralleling pandemic states such as SARS-CoV and MERS-CoV, [18] provide effective lessons for adequate OHSM in dealing with COVID-19 in the healthcare setting. 

### 1.1. Current State of Research

Results from previous studies conducted after the SARS (2003) [19,20] and influenza (2009) [21] pandemics showed that wearing personal protective equipment (PPE) proved to be a relevant measure to protect healthcare professionals from infections while working with patients, especially regarding its quality, fitting, compliance in use and training measures [20]. However, stockpiling of PPE as well as training on infection control measures were lacking then [10]. Moreover, the risk of indirect virus transmission, e.g., via contaminated surfaces was addressed by hand hygiene, intense hygiene regulations and surface disinfection [22,23]. Furthermore, a reduction of activities generating virus-rich aerosols to a minimum, adequate ventilation, quarantine and isolation of infected, was considered important for infection prevention [20]. Daily (self-) monitoring of healthcare professionals for symptoms of infection also proved useful to prevent spreading events at the workplace [10]. In addition the inclusion of organisational (e.g., availability of PPE) and behavioural measures (e.g., individual training) as OHSM in the healthcare setting during the SARS pandemic led to better precautions against the pandemic of healthcare workers [24]. A clear, regular communication and the provision of up-to-date pandemic-related information via digital means proved to be profitable [25]. Increased stress levels due to higher workload and feelings of anxiety because of the risk of infection were addressed by psychosocial support of the employer [26] or even by professional psychological counselling on site [25].

Preliminary results on OHSM during the COVID-19 pandemic show great similarities in protective measures to previous pandemics affecting the respiratory system. Recent studies have shown that the use of PPE consisting of disposable medical gloves, foot coverings, gowns, hair and head coverings, eye protection, hand disinfection as well as mouth nasal protection (surgical-mouth-nose protection, high-quality filtering face piece (FFP) masks [27]) [28,29,30,31] is still recommended when working in close patient contact. FFP masks have a higher protective effect than everyday masks for private use (sewn from commercial fabrics, no filter performance). FFP masks are items of personal protective PPE within the scope of occupational health and safety. They protect the wearer of the mask from particles, droplets and aerosols and are advised to be used in risky occupations as they offer external and self-protection. For a different class of protection, they are divided into FFP2 and FFP3 masks. FFP2 masks must filter at least 94% and FFP3 masks at least 99% of aerosols. In contrast, a surgical mask, for example, only offers external protection (protection from droplets, low protection from aerosols) which is why FFP masks are important in terms of PPE [32]. The national pandemic plan drawn up by the Robert Koch-Institut [27] in Germany clarifies the relevance of PPE of outpatient workers, such as gloves, FFP masks and protective gowns for influenza prevention. Governmental guidelines in terms of the outbreak of the COVID-19 pandemic emphasizes the use of PPE and FFP masks for outpatient caregivers while caring for their patients [33]. Additionally, it is recommended that patients should also wear a mask when outpatient care service is provided [33,34]. However, a short-term shortage of PPE which was reported across countries, was aggravating infection prevention in outpatient and inpatient settings, particularly during the initial peak of the COVID-19 pandemic in March 2020 [9,35,36,37]. Nevertheless, this fact again highlights the importance of stockpiling PPE in crisis situations [38,39,40]. Moreover, training is also recommended for the correct use of PPE, since contaminated PPE poses a risk of transmission of the virus, e.g., while donning and doffing [41]. Outpatient caregivers should also be able to maintain the recommended safe distance in the patient’s home, and rooms used should be adequately ventilated for the prevention of indirect virus transmission [42]. In addition, indirect virus transmission must be avoided by hygiene interventions such as surface disinfection, as well as by the adaption of behaviour, e.g., via extensive hand hygiene and avoidance of hand-to-face contact [30]. Therefore, service vehicles and other work equipment of outpatient caregivers should also be regularly disinfected and preferably only used by one person. Before entering the service rooms after care, the outpatient caregivers should also decontaminate themselves. If possible, office work should be carried out in the home office [42]. Furthermore, internal crisis management necessarily includes a pandemic plan [38]. Guidelines for the development of such an internal plan can be found in the Robert Koch-Institut (RKI’s) national pandemic plan [27]. 

In summary, several studies and recommendations for action can already be found with regard to the use of OHSM in the healthcare sector. Although there are recommendations for action on OHSM for COVID-19 for outpatient care [33,34,42], there are no studies describing implemented OHSM in outpatient care services in Germany yet. Furthermore, the needs and wishes of German outpatient caregivers regarding OHSM especially during pandemic situations also have not been examined so far. However, in terms of the German Occupational Health and Safety Act (ArbSchG) [43] and the great systemic relevance of outpatient caregivers, it is of crucial importance to identify the existing OHSM, as well as the corresponding needs for improvement in OHSM to ensure their health and wellbeing during this pandemic state.

### 1.2. Study Aims and Research Questions

The aim of this study was to investigate the OHSM that have been implemented in outpatient care in Germany since the outbreak of the COVID-19 pandemic and to identify related occupational health and safety deficiencies. In addition, the needs of outpatient caregivers with regard to OHSM in their work setting were examined.

We proposed the following research questions:What specific OHSM have been implemented in outpatient care since the outbreak of the COVID-19 pandemic?What are identifiable unmet needs and wishes of outpatient caregivers with regard to OHSM considering the COVID-19 pandemic?

## 2. Materials and Methods

### 2.1. Participant Selection and Interview Conduct

Semi-structured telephone interviews (*n* = 15) were conducted with outpatient caregivers from outpatient services in Northern Germany. To contain the incidence of infection during the data collection period it was decided to conduct telephone interviews only to avoid face-to-face contacts, as it was also stipulated by the government [14]. The interviews were carried out in May 2020 and June 2020. Inclusion criteria for the recruitment of outpatient caregivers implied work experience in outpatient care of more than 6 months, performance of the work activity in Hamburg, and fluency in the German language. Contact with the outpatient care services and possible respondents was made via invitation emails, telephone calls and via social media. The shortest interview endured 26 min, the longest was about 60 min. All quotations used for the publication were translated from German into English.

### 2.2. Interview Guideline

Based on a prior literature review, a semi-structured interview guideline was designed [44]. The questions were critically reviewed with reference back to the research theme, sorted according to chronological order and subsumed into categories [45]. A pre-test interview was performed to improve the interview guidelines where applicable. All telephone interviews were conducted by the main researcher (NM). An extract of the interview guidelines is shown in Table 1.

### 2.3. Analysis

In this present study, the subjective perceptions of outpatient caregivers with regards to pandemic-related OHSM in their settings, as well as their opinions on unmet needs and wishes regarding OHSM, were the focus of the analysis [46]. The telephone interviews were tape-recorded for verbatim transcription by research assistants according to Kuckartz [44]. For data protection reasons, the transcripts were anonymised. The interviews were then analysed using Mayring [47] qualitative content analysis in a deductive-inductive procedure within MAXQDA 2020 (VERBI Software, 2019, VERBI GmbH, Berlin, Germany). During analysis, the principal investigator identified and refined codes categories and subcategories in an iterative process. This process was assisted by colleagues from the research team to achieve accuracy and consensus on the coding system. The final coding system was summarised in another separate document in which the material was further reduced and compacted by two members of the research team. Throughout the analysis process, reflexivity and transparency were maintained in relation to the potential influence of the researchers’ goals and biases on the findings as well as on the interpretations.

## 3. Results

### 3.1. Sample Characteristiccs

The total number of the sample was *n* = 15, of which 12 were female respondents. The age ranged from 21 to 67 years (as shown in Table 2). Thirteen of the interviewed outpatient caregivers worked full-time with work experience ranging from seven months to 36 years. Seven respondents had a secondary school leaving certificate, four had a general university entrance qualification, one had a technical college entrance qualification and one had an adult education entrance qualification. Most of the 15 respondents were qualified as geriatric nurses.

### 3.2. Identifiable Occupational Health and Safety Measures (OHSM) in Outpatient Care Services

Interviewed outpatient caregivers described several OHSM which they perceived being implemented in their workplaces. Within this framework, supply and use of PPE for infection prevention, consultation of company doctor and occupational safety specialist, social distancing, availability of an in-house pandemic plan, recommendations, advice and obligations, education and training, possibility to refuse care services for COVID-19 positive patients, paid recreational leave, information transmission and perceived level of information of the employer were identified as subcategories.

#### 3.2.1. Supply and Use of Personal Protective Equipment (PPE) for Infection Prevention

Most of the interviewed outpatient caregivers had already used PPE in their daily work before the COVID-19 pandemic. However, the usage of PPE became more important in the course of the pandemic. Some interviewees reported an initial shortage of PPE in the first peak of infections caused by the coronavirus in March 2020, which resulted in fear and stress because of the risk of infection. However, the shortage of PPE was remedied in the course of time and a sufficient supply of PPE was achieved for nearly all outpatient caregivers surveyed. PPE used included simple mouth-nose-protection, certified and more effective FFP1- and FFP2-masks, gowns, body suits, gloves and hand disinfectants.

“*(...) This care facility already had the protective masks before the outbreak (...).*” (Interviewee #2)

“*(...) it was stressful that there was not enough protective equipment, i.e., disinfectants, FFP2-masks and so on, it was quite exhausting at that time. I was afraid, how could I protect myself (...). But in the meantime the deliveries are coming in little by little, and now I look at things more calmly.*” (Interviewee #6)

Furthermore, it was mentioned that PPE was also provided to patients to reduce the risk of infection for both sides, the caregiver and the patient.

“*(...) We have handed out washable face masks to our customers in order to protect ourselves once more during the care process, double protection so to speak (...)*” (Interviewee #11)

#### 3.2.2. Consultation of Company Doctor and Occupational Safety Specialist

Consultation by a company doctor or an occupational safety specialist, e.g., in terms of asking questions concerning the coronavirus, was possible for almost all outpatient caregivers in the present sample.

“*(...) Yes, of course, we can reach him by phone or in the meantime they also offer video calls.*” (Interviewee #7)

The possibility of a free in-house test by the company doctor, in case of suspected COVID-19 infection, was also given.

“*(...) we can get the test if we want it. I think/I think it was 60 Euro or something like that, so we have this cooperating doctor, who has enough of the tests and who provides us with it and we unfortunately have to pay her privately, unless something is suspected.*” (Interviewee #8)

Only a few interviewees stated that neither a company doctor, nor an occupational safety specialist was attached to their care service so an individual consultation was not possible. Others made clear that offers were not taken or the availability of an occupational safety specialist was unknown.

“*Yes, we have a company doctor. Unfortunately, he is not allowed to come. (...). No, I wouldn’t know. No, none for occupational safety.*” (Interviewee #14)

“*Yes, we have a company doctor. Actually we do, but none of us were actually there yet, because this was when Corona pandemic broke out.*” (Interviewee #1)

#### 3.2.3. Social Distancing

Furthermore, great attention was paid to social distancing in some outpatient care services. Occasionally, more distance was maintained while taking care of patients, however, without reducing the quality of care, e.g., in basic care services.

“*Avoiding contact, keeping more than one and a half metre distance, if it is possible, it is not always possible due to basic care or when we put on compression stockings (...)*” (Interviewee #9)

In the course of social distancing, it was also reported that team meetings were reduced or abolished by the management.

“*Team meetings have been cancelled in order to keep the/because of the minimum distance and infection reduction.*” (Interviewee #11)

#### 3.2.4. Availability of an In-House Pandemic Plan

Few interview partners were told about the existence or development of an in-house pandemic plan. Besides, pandemic plans mentioned already existed before the COVID-19 outbreak and were not specially adjusted to COVID-19.

“*(...) we have a pandemic plan, it is independent of Corona, it is simply valid if any pandemic occurs and there are (very clear) rules in it (...)*” (Interviewee #4)

#### 3.2.5. Recommendations, Advice and Obligations

Moreover, recommendations, advices and obligations were made to the interviewees by their employer. On one occasion, the use of public transport was explicitly discouraged by the employer to reduce the risk of infection. 

“*(...) There are higher-level instructions, yes. So/so to use public transport less (...).*” (Interviewee #4)

In addition, a company doctor recommended outpatient caregivers not to get vaccinated against other infectious diseases (e.g., flu) to prevent weakening of the body’s own defences, which could increase the susceptibility to infection with COVID-19.

“*We were advised doing this later because we would not know what it would be like if we were to be infected with corona and if we were going through some other immunisations right now, we would be doubly weakened. We should then put that on hold.*” (Interviewee #11)

Employees were also asked to continuously check upon their clients for symptoms of a COVID-19 infection (fever, loss of smell and taste, coughing etc.)

“*And, of course, what we also do for clients is take their temperature every day, every day before we do anything.*” (Interviewee #9)

#### 3.2.6. Education and Training

Training measures, e.g., on hygiene regulations, were described as the basis for infection prevention. It was noticed that knowledge would now be increasingly reminded, referred to and controlled by the employer.

“*In other words, through further education and training? Well, that’s big now. It is also pointed out again and again. Employees are always being trained in some way or another. And they also check whether their hands are properly disinfected. These are small things that you have learned in training, but many of them are somehow blunted and this is now being called up again.*” (Interviewee #2)

#### 3.2.7. Possibility to Refuse Care Services for COVID-19 Positive Patients

In one outpatient care service, employees had the option of refusing corona-positive patient care in order to protect themselves against COVID-19 infection.

“*(...) if a patient is tested positive, we are allowed to refuse care, one of our employees was pregnant for example, or had small children.*” (Interviewee #10)

#### 3.2.8. Paid Recreational Leave

Furthermore, an outpatient care service offered paid time off for recreation and regeneration to its employees to compensate the additional occupational burden of the COVID-19 pandemic.

“*(...) our outpatient care service has organised this very well (...) so that someone with stress due to the virus can recover. We got time off that was nice.*” (Interviewee #10)

#### 3.2.9. Information Transmission

The majority of respondents were informed about COVID-19-related news in person, either face-to-face or in team meetings. Written notes, information brochures and electronic media, e.g., telephone, email, social networks or internal online platforms, were also frequently used by employers to transmit information. Only a few employees were informed by post. In one case receipt of information was also confirmed by a receipt.

“*This is mainly done via notes and otherwise people get it personally. So we have compartments where the information is stored. And if it concerns someone personally, they will be called.*” (Interviewee #8)

#### 3.2.10. Perceived Level to Which the Employer Is Informed

The majority of all respondents felt that the level of information about the current state of knowledge regarding COVID-19 and the adequate handling of the disease was good or very good. In contrast, a few people perceived their employer’s level of information as mediocre or differentiated.

Some of the outpatient caregivers highlighted the effectiveness and frequency of information by the employer. The answers given were differentiated. Some employees assessed it as sufficient, helpful or even sporadic.

“*There have always been good briefings from the management when new hygiene regulations were introduced. We have notices everywhere regarding Covid-19 from the RKI. I think that’s very good.*” (Interviewee #11)

“*No, not regularly. This was done once, because we do not have the time. We are so short of time and then there is enough time/then it is not done that way, right?*” (Interviewee #14)

Table 3 below is intended to provide an overview of the identified OHSMs and the number of outpatient caregivers who experience them in the course of their work.

### 3.3. Unmet Needs and Deficiencies Regarding OHSM in Outpatient Care Services

Deficits in the perceived OHSM and unmet needs of the ambulant caregivers with regard to OHSM emerged from the interviews. The subcategories satisfaction with OHSM, analogue information channels only and perceived absence of OHSM were identified.

#### 3.3.1. Satisfaction with OHSM

All in all, entire satisfaction with the OHSM offered was mentioned only by a few interviewees.

“*All precautions have been taken and everyone is accordingly motivated to implement them in the same way.*” (Interviewee #9)

#### 3.3.2. Analogue Information Channels Only

Shortcomings in OHSM became apparent during the interviews with outpatient caregivers. For instance, the shortcoming was due to the limitation of OHSM to written information material on the pandemic only.

“*I didn’t notice anything, except that information flyers are now displayed here.*” (Interviewee #12)

#### 3.3.3. Perceived Absence of OHSM

Furthermore, some outpatient caregivers experienced no occupational health and safety measures at all in their care services.

“*No special protection/No. Work is being done, because the work is always coming up, that’s where you have to go.*” (Interviewee #9)

### 3.4. Wishes of Outpatient Caregivers Regarding OHSM

The ambulant carers expressed their wishes in relation to OHSM in their workplace, thus financial and social recognition, requirements for PPE, provision of home office, Development of in-house pandemic plans, coverage of the need for information and work organisation were identified as subcategories.

#### 3.4.1. Financial and Social Recognition

Qualitative data analysis revealed that the most frequent request in the context of recognition at work was the receipt of financial recognition for the professional challenges of the pandemic. Furthermore, more social recognition for the professional challenges of the pandemic was also requested. Apart from that, more social recognition from the manager for the professional challenges of the pandemic was also once demanded. 

“*I want to have a general social recognition and financial compensation that makes my job worth doing. We bear so much responsibility. We work towards the doctors. We make pre-diagnostics, or rather, we are the ones who first notice the symptoms and react, and then we get paid so little (...)*” (Interviewee #11)

“*It’s kind of weighing a bit more on the shoulders now. A bit more responsibility too. And I think that after that you should do something good for the employees as a manager. You have to see how it turns out.*” (Interviewee #2)

#### 3.4.2. Requirements of PPE

Need for improvement was partially also expressed with regard to the qualitative and quantitative supply of PPE, hygiene measures and the provision of testing for medical professionals.

“*Actually I should have an FFP2, 3 mask (...) So, the only thing I wish for in the future is for this situation to never happen again, there are shortages with mouth-nose-protection or hand disinfectant. (…) I consider this access for medical staff for testing simply stupid. Because if I stay at home for 14 days because there is no testing, I will miss 14 days. But if a test is taken and after three days it says: “Hey, it’s negative, have fun! I could go back to work.*” (Interviewee #11)

#### 3.4.3. Provision for Working from Home

Further room for improvement on health promotion and occupational health and safety included the consideration to offer the possibility of doing home office work. 

“*I sometimes envy the people who sit in the home office. Where I say it’s a pity, I would love to work at home and then be able to control my children who sit at home and can’t go to school. But I have to work while the neighbours all around are allowed to sit at home.*” (Interviewee #12)

#### 3.4.4. Development of In-House Pandemic Plans

The development of a guiding in-house pandemic plan was considered as helpful to structure a course of action in dealing with the challenges of the pandemic.

“*Yes, well, in principle that would be a kind of pandemic plan (...) but nobody has that, right? That you know in principle, how it works exactly, that you roughly know in which direction it goes, but you don’t know anything more precise.*” (Interviewee #6)

#### 3.4.5. Coverage for the Need for Information

Correspondingly, it was also stressed that the need for information about the new situation and challenges of the pandemic was not yet sufficiently covered.

“*So of course I would like it to be as soon as there are some changes in status somewhere (...) that one is actually always informed regularly.*” (Interviewee #6)

#### 3.4.6. Work Organization

With regard to risk groups in the team, it was mentioned that personal risk factors, e.g., pre-existing conditions, higher age or pregnancy, should be considered more in work organisation. These risk groups should also receive more social support.

“*I think that, in general, it should be possible to give workers at risk time off and that they should also receive social support in this context. That you can better protect your colleagues by simply allowing them to take a complete leave of absence without having to face any consequences in terms of labour law or whatever else.*” (Interviewee #13)

Table 4 sums up identified unmet needs, deficiencies and wishes reported by outpatient care service regarding OHSM during the COVID-19 pandemic.

## 4. Discussion

### 4.1. Discussion of the Interview Results

By conducting our qualitative interview study, we were able to gain insights into the pandemic-related OHSM in the present outpatient care services, as well as into the deficiencies and the perceived needs regarding OHSM of the surveyed outpatient caregivers. Due to the sudden outbreak of the coronavirus pandemic, tailored OHSM had to be implemented or existing OHSM had to be adapted to the new challenges posed by the pandemic. Outpatient caregivers reported that the usage of PPE became more important during the coronavirus pandemic and that they had to face short-time shortages of PPE in the first peak of the pandemic in March 2020. However, a sufficient supply with PPE was achieved for nearly all outpatient caregivers surveyed. These results were also strongly reflected in the current literature which reported on the importance of PPE in the healthcare sector [28,29,30,31] and of nationwide shortages in its supply [9,35,36,37]. In most outpatient care services, the respondents were able to seek advice and help from a company doctor or occupational safety specialist. However, gaps in this offer were identifiable, as their consultancy was partly not used or contradictory recommendations were given to the participants of the study, e.g., in discouraging important vaccinations against other infectious diseases such as influenza. Due to new hygiene and governmental regulations, working practices and related OHSM had to be adjusted. In our study, the majority of outpatient caregivers perceived the effectiveness and frequency of information by their employer, as well as the level to which they were informed, as good or very good. However, the means of exchange of information varied in the care services. In some cases, analogue information channels were used only. In-house pandemic plans existed rarely, even though they are essential with regard to internal crisis management [38]. However, where they existed, the plans were not specifically tailored to COVID-19 because they existed before the outbreak of the coronavirus pandemic. Furthermore, several measures were rarely mentioned or were specifically related to the environment. Paid leave to compensate for the occupational challenges of the pandemic, for example, was only once offered in our sample. Moreover, the possibility of rejecting care for COVID positive patients was only reported once, forcing outpatient caregivers into a very high risk position. Furthermore, few of the ambulatory caregivers mentioned pandemic-related training interventions, although training seems important in these times for dealing with the challenges of the pandemic, e.g., in adequate use of PPE [41]. All in all, with all that is going well, it should also be emphasised that entire satisfaction with the OHSM offered was rarely mentioned and that some outpatient caregivers even perceived an absence of OHSM in their setting. Accordingly, there is necessarily need for action to improve the implemented OHSM. 

The wishes and needs of the study participants with regard to OHSM in their setting were broadly diversified and very individual. This may be explained by the different circumstances in the care services, as well as the individual requirements and focuses in terms of work and health of the outpatient caregivers. Few statements revealed that the qualitative supply of PPE was still lacking in FFP-Masks. Nevertheless, most of the interviewees reported that this shortcoming was overcome and that they were well supplied with personal protective equipment for their personal protection in their daily working life. 

Lack of financial reimbursement and social recognition in the caregiving profession has been an ongoing issue for years and is discussed as a factor for career retention [48]. Correspondingly, outpatient caregivers emphasised that from both a financial and social perspective, more recognition was desired for the professional challenges of the pandemic [38]. Given the fast-moving nature of the news on COVID-19, the need for improvement in the information provided by employers was also requested. Finally, the possibility of doing home office work was requested, as much as realisable. However, digital tools (soft- and hardware) were mentioned as a missing resource in this context. Moreover, it was desired to adjust the organisation of work with regard to personal risks factors of an infection. In doing so, these at-risk groups should also receive social support by their employer and their colleagues [49].

### 4.2. Strengths and Limitations

A strength of our study is the fact that we were able to recruit a solid number of outpatient caregivers from Northern Germany with different socio-demographic characteristics and different workplaces at very short notice after the outbreak of the coronavirus pandemic. Thus, we were not only able to present a very broad picture of opinions regarding OHSM, but also, due to the short-term nature of the data collection, a very up-to-date picture of opinions, need for improvements, and requests of the outpatient caregivers regarding OHSM. The results provide answers in a field of research that has not been explored before. By describing our findings in detail and substantiating the research results with the help of direct quotes from the interviewees, the results could be presented in a trustworthy way. In addition, the research findings were discussed in depth in a research group and contrasted with empirical evidence.

As limiting factors of our study, it should be noted that our study comprised a relatively small sample size, therefore results need to be reconsidered in terms of transferability and generalisability. Therefore, further control studies would be useful [46]. However, the results are probably difficult to transfer and to generalize anyway, as outpatient care services are often privately owned small and medium-sized enterprises [13]. Furthermore, results are based on a random sample, recruited according to the snowball principle [46]. Self-selection of participants, also with regard to interest in participating in the study, can, therefore, not be ruled out. Furthermore, our sample shows a surplus of women, which may also be due to the fact that more women work in outpatient care in Germany [13]. Another methodological limitation is that personal interaction with the respondents by means of facial expressions and gestures was not given in the telephone interviews. In terms of the interaction and communication situation [50], this could have caused deductions in the relationship of trust and mutual understanding between interviewee and interviewer [45].

### 4.3. Practical Implications and Recommendations for Further Practice and Research

Further research studies with larger sample sizes would be necessary to evaluate transferability and to substantiate our study results, since the work setting of outpatient caregivers represents a special group among caregivers due to its special characteristics. Considering that the data collection of our study took place at the beginning of the pandemic, a follow-up study might provide information on changes, adaptations and lessons learned with regard to pandemic-related OHSM in the course of time. Furthermore, outpatient care services are predominantly small and medium-sized enterprises [13], therefore customization of OHSM with regard to the individual needs of the outpatient care services, as well as of its staff and its patients are necessary. However, it must be noted that the hierarchy of hazard control was primarily developed for controlling chemical hazards [51], thus OHSM addressing COVID-19 face the challenge that health risks due to the virus are more difficult to reduce by technical or organisational measures. Therefore, measures of personal protection are especially relevant and must be constantly available, particularly effective and safely applied. In order to provide employees with adequate personal protection at all times, a well thought-out internal storage system for PPE might be recommendable [40]. In this context, management systems for stockpiling with PPE in healthcare have been successfully tested in first pilot projects [52]. The implementation of such systems promises to enable a more economical, efficient use of PPE in case of shortages via forecasting or to even avoid future shortages of PPE altogether [52,53]. In addition, the correct selection of the adequate PPE, depending on the activity on the patient, has to be considered [53,54]. Last but not least, training interventions focusing on the adequate use of PPE and infection-related self-protection should be implemented and regularly refreshed [41,55]. However, further research is needed to foster compliance in the implementation of pandemic-related interventions taught in the training, as well as in the communication strategies used by healthcare services to improve occupational safety in the setting [24,56]. To achieve a representative study sample size should be expanded taking into account various characteristics, e.g., different ages or a gender distribution by using a quantitative questionnaire study.

Moreover, the development and implementation of internal pandemic plans for an improved, structured handling of the challenges posed by the pandemic in everyday work in outpatient care would be advisable [27,38]. In order to prevent uncertainty and fear among employees regarding information about COVID-19, pro-active communication about general behaviour at work, infection prevention, disease characteristics and its possible health risks, is essential. Therefore, measures and clear communication must be undertaken as early as possible [57]. In this context, the provision of analogue information seems to be outdated for the transmission of information. Thus, the promotion of digitisation in outpatient care also needs to be addressed [9,38]. Moreover, increased publicity and policy work is needed to improve the perceived financial and social recognition of ambulatory caregivers [9,38]. Furthermore, several studies [10,58] and our study results indicate psychosocial stress in health care professionals caused by the pandemic. Therefore, a study focusing on the OHSM in relation to mental stress of outpatient caregivers would be useful. Nevertheless, offering psychological support to staff in times of crisis is generally recommended [25,59]. 

In summary, to be able to implement the recommendations mentioned in a unified and effective manner, it might be recommendable to implement multi-headed occupational health management in each outpatient care service. Being a division of the company, occupational health management is enabled to better customize OHSM at the administrative and behavioural level. Furthermore, it can constantly monitor, evaluate and adapt the internal OHSM just in time for the dynamic development of the pandemic. Moreover, the provision of employees with more individual OHSM, taking into account their personal needs, could be enabled. All in all, this could make it a central point of contact regarding OHSM, supporting outpatient caregivers in everyday work during the coronavirus pandemic.

## 5. Conclusions

This qualitative study is the first to examine the pandemic-related OHSM in German outpatient care services during the COVID-19 pandemic with regard to their status quo, existing potential for improvement and in relation to the needs of outpatient caregivers themselves. Our respondents reported on a large number of OHSMs in their care services that were implemented or adjusted in the course of the pandemic. These included in particular the use of PPE and infection control regulations. In the first peak of the pandemic, hurdles in OHSM had to be dealt with, such as serious shortages in PPE supply. In addition, outpatient caregivers expressed wishes for more financial and social recognition for the occupational challenges of the pandemic, as well as for improvement with regard to the qualitative and quantitative provision of PPE. Our findings suggest that there is potential for improvement in the already implemented OHSM, which could be addressed with the help of our recommendations. All in all, the lessons from previous pandemics regarding OHSM and our results should provide a suitable basis for the development and improvement of specific OHSM against COVID-19 and for possible future infectious disease pandemics to protect outpatient caregivers’ health and well-being at work.

## Figures and Tables

**Table 1 ijerph-18-02987-t001:** Interview topic list.

Phase 1	Study information, confidentiality, informed consent
Phase 2	Qualifications, working activity
Phase 3	Available occupational health and safety measures (OHSM), further requests and needs
Phase 4	Socio-demographics of the interviewees and farewell

**Table 2 ijerph-18-02987-t002:** Participant characteristics (*n* = 15).

ID	Gender	Age	Date of Interview	Qualification	Occupation
1	f	31	7 May 2020	Caregiver	Outpatient geriatric nurse
2	f	31	7 May 2020	Geriatric nurse	Outpatient geriatric nurse
3	f	33	7 May 2020	Geriatric nurse	Outpatient geriatric nurse
4	m	64	8 May 2020	Geriatric nurse	Outpatient geriatric nurse
5	f	21	12 May 2020	Home and family care	Outpatient home and family caregiver
6	m	51	12 May 2020	Geriatric nurse	Outpatient geriatric nurse
7	m	25	15 May 2020	Geriatric nurse	Outpatient geriatric nurse
8	f	38	15 May 2020	Healthcare and nursing staff	Outpatient caregiver
9	f	51	19 May 2020	Geriatric nurse	Outpatient geriatric nurse and office manager in health sector
10	f	36	3 June 2020	Social manager	Outpatient caregiver
11	f	46	11 June 2020	Geriatric nurse, additional qualification intensive and palliative care	Outpatient geriatric nurse
12	f	50	11 June 2020	Wound expert	Care specialist and nutrition manager in the outpatient care
13	f	34	15 June 2020	Geriatric nurse	Care specialist and deputy care management in the outpatient care
14	f	67	19 June 2020	Geriatric nurse	Outpatient geriatric nurse
15	f	37	29 June 2020	Geriatric nurse and wound expert	Outpatient geriatric nurse and wound expert

f = female, m = male.

**Table 3 ijerph-18-02987-t003:** Identified OHSM mentioned by interviewed outpatient caregivers.

Identified OHSM	Outpatient Caregivers (*n*)
Supply of PPE ○ Mouth-nose-protection ○ Filtering face piece (FFP) – FFP1/FFP2-masks ○ Gowns ○ Bodysuits ○ Gloves ○ Disinfectants	11
Supply of PPE for patients	1
Consultation of company doctor	12
Free in-house coronavirus test by company doctor	1
Consultation of occupational safety specialist	8
Keeping distance by cancellations of personal team meetings	1
In-house pandemic plan available	2
Supervision of employer ○ Recommendations ○ Obligations	2 2
Advice from company doctor	1
Education and training	1
Possibility to refuse caring for coronavirus-infected patients	1
Paid recreational leave	1
Perceived level to which the employer is informed	13

**Table 4 ijerph-18-02987-t004:** Unmet needs, identifiable deficiencies and further requests in terms of OHSM of outpatient care services.

Unmet Needs and Identified Deficiencies	Requests
Shortage of PPE, especially in the beginning of the outbreak	Supply of qualitative and quantitative PPE
No company doctor/occupational safety specialist	Possibility of testing
Information by employer was not on a regular basis	Financial and social recognition
Absence of OHSM	Appreciation by employer
OHSM limited to written material	Possibility of working from home
Lack of information	Development of pandemic plans
	Higher information transmission
	Higher consideration of personal conditions while working, e.g., pregnancy
	Social support

## Data Availability

The data analysed during the current study are not publicly available due to German national data protection regulation. They are available on individual request from the corresponding author.

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
