# Peer review of "Occupational Health and Safety Measures in German Outpatient Care Services during the COVID-19 Pandemic: A Qualitative Study"

_ijerph, 2021, doi:10.3390/ijerph18062987_

Round 1
Reviewer 1 Report
The objectives of this paper were to characterize occupational health safety measures (OHSM) implemented in the outpatient care settings during the COVID-19 pandemic and identify limitations of the OHSMs. This is an interesting topic that provides an important understanding of OHSMs currently implemented and their effectiveness perceived by healthcare workers. This study used a qualitative research method for data collection. I have some specific comments below.
Introduction
- The current state of research (Page 2, line 72): I think the authors should cut the section summarizing the literature on the SARS outbreak and influenza pandemic to 3 to 5 sentences. It is more important to emphasize current OHSMs implemented in outpatient settings during the pandemic which the authors described in the following paragraph on page 3.
- Page 3, Line 100: I recommend mentioning the government PPE ensemble recommendations for healthcare workers when providing outpatient care during the COVID pandemic.
- Page 3, Line 104: Write out the full name of FFP. This is the first time this acronym is mentioned in the text. It is important to distinguish the difference between a respirator (e.g., N95) and a surgical mask.
Results
- Page 6, Line 208: What are the FFP1 and FFP2 masks? It would be helpful to clarify the respirator types here.
Author Response
Reviewer 1
The objectives of this paper were to characterize occupational health safety measures (OHSM) implemented in the outpatient care settings during the COVID-19 pandemic and identify limitations of the OHSMs. This is an interesting topic that provides an important understanding of OHSMs currently implemented and their effectiveness perceived by healthcare workers. This study used a qualitative research method for data collection. I have some specific comments below.
Introduction
- The current state of research (Page 2, line 72): I think the authors should cut the section summarizing the literature on the SARS outbreak and influenza pandemic to 3 to 5 sentences. It is more important to emphasize current OHSMs implemented in outpatient settings during the pandemic which the authors described in the following paragraph on page 3.
Thank you very much for your feedback. We have shortened the paragraph. However, we would like to keep the rest of the content as background information.
- Page 3, Line 100: I recommend mentioning the government PPE ensemble recommendations for healthcare workers when providing outpatient care during the COVID pandemic.
Thank you; we added the missing information by mentioning the national pandemic plan and the additional information during the COVID-19 pandemic.
- Page 3, Line 104: Write out the full name of FFP. This is the first time this acronym is mentioned in the text. It is important to distinguish the difference between a respirator (e.g., N95) and a surgical mask.
Thank you for pointing this out. In the penultimate paragraph of the current state of research, we explained FFP masks and clarified the difference to a surgical mask.
Results
- Page 6, Line 208: What are the FFP1 and FFP2 masks? It would be helpful to clarify the respirator types here.
Thank you for pointing this out. In the penultimate paragraph of the current state of research, we have explained the differences between the FFP masks (after we first introduced them) and hope that this will suffice for the rest of the article.
Reviewer 2 Report
See attached file

Author Response
Occupational health and safety measures in German outpatient care services during the COVID-19 pandemic: a qualitative study Congratulations for your article. This article addresses an interesting topic refers to a minority health sector who needs a correct hygiene and safety measures. Reviewing the bibliography, no similar articles have been published, so it is necessary to show the problems and needs that other health sectors may have. The article is complete and well structured. The title is concise, clear and attractive. The summary shows the content of the article, with adequate and formal language. The introduction explains the background and acquaint the reader with the study problem with bibliographic references in an appropriate way. It also clearly formulates the objectives of the study. The study methodology is correct. The main limitation, as the authors have reflected, is the small number of study participants (n = 15). This is a descriptive study and statistical analysis is lacking, although it is possible that the small number of participants made it difficult to obtain statistically significant results. The results are in accordance with the objectives. However, I believe that a summary table that specifies the identifiable OHSM in outpatient care services as well as the number of participants who acknowledge having received them could improve the article. In the same way, it could be interesting to make a summary table with the unmet needs and identifiable deficiencies related to security measures in the covid 19 pandemic. The discussion and conclusions are adequate. The discussion responds to the stated objectives and compares the results with the most recent and relevant ones published at the moment. It reflects the main limitation of the study while proposing the need to continue studying in this area. The bibliography is adequate, extensive and up-to-date.
Thank you very much for your feedback.
- Correct, the small number of participants made it difficult to do further statistical analysis. We have therefore added the reference to a quantitative study in the implications for further research.
- Thank you very much for the advice. We have supplemented the tables (Table 3 and Table 4) in the results section accordingly.
- “Table 3: Identified OHSM mentioned by interviewed outpatient caregivers.” (Page 4)
- “Table 4: Unmet needs, identifiable deficiencies and further requests in terms of OHSM of outpatient care services.” (Page 6)